# A Dual-Band Dual-Antenna System with Common-Metal Rim for Smartphone Applications

**Ziqiang Xu [1],[*]** , **Chenguang Ding [1]**, **Qiangqiang Zhou [1]**, **Yangtao Sun [1]** and **Si Huang [2]**

1    School of Materials and Energy, University of Electronic Science and Technology of China, Chengdu 611731, China; ding_163_1@163.com (C.D.); h_xiasummer@163.com (Q.Z.); zz_filter@hotmail.com (Y.S.)
2    Department of Electrical Engineering, University of Arkansas, Fayetteville, AR 72701, USA; sxh039@email.uark.edu
*    Correspondence: nanterxu@uestc.edu.cn

**Abstract:** A dual-antenna system operating at WIFI and GPS bands is proposed for common-metal rimmed smartphones applications. This dual-antenna system, which is horizontally placed on a ground plane of $4.5 \times 75$ mm$^2$, consists of two folded inverted-F antennas (IFAs) sharing the same metal rim. Each IFA contains part of the metal radiating arm, and both IFAs own approximately one-quarter free space wavelengths at 2.44 GHz. A matching network is embedded in the feeding line of the left IFA to provide a resonant frequency at 1.575 GHz. By adjusting the positions of the shorting branch and feeding line, good impedance matching is obtained. Two gaps in the metal frame and a center shorting branch between two IFAs are adopted to improve the isolation. The isolations of better than 22 dB and 19 dB in GPS and WIFI bands are attained, respectively. The proposed antenna is fabricated, and the measured results regarding S-parameters, radiation efficiency, gain, as well as diversity performances are presented.

**Keywords:** dual-antenna system; WIFI and GPS bands; common-metal rim; matching network

## 1. Introduction

The metal-rimmed smartphone has become a very popular electronic device in recent years, owing to its fashionable, aesthetic appearance, and enhanced mechanical strength. However, the metal rim weakens the radiation performance of the antenna inside the smartphone and causes undesired mutual coupling. To solve these problems, some solutions were provided successively in references [1–5]. An inverted-F antenna for the smartphone surrounded by a metal rim with two gaps is proposed in reference [1]. By cutting through the metal rim, the radiation performance of the antenna can be significantly improved. Moreover, with the increasing demands of heterogeneous functions of mobile terminals, smart phones also need to transmit a large amounts of mobile data, including voice, video, picture and so on. Compared to mobile data traffic, WIFI has a faster transfer speed at a lower price, so it becomes top priority while transmitting massive data for terminal devices [6,7]. To further realize a higher data rate without additional power, multiple-input multiple-output (MIMO) antennas with proper isolation are being widely applied in mobile terminals as well [8–14]. However, for the metal-rimmed smartphone utilizing multiple antennas, the high mutual coupling in smartphones will occur inevitably when the MIMO antennas share the common part of metal rim and work in the same frequency band. Consequently, how to solve this problem becomes a serious technical challenge for designers. Several decoupling methods were reported, including neutralization line (NL) technology [15], a parasitic element [16], defected ground structure (DGS) [17], etc. For example, in reference [17], a defected ground plane combined with an inverted T-shaped slot stub and rectangular

slot ring is used to reduce the mutual coupling between two MIMO antenna elements. Yet this technique is not suitable for multiple antennas with common-metal rim applications.

Besides, with the rapid development of electronic communication, the global position system (GPS) integrated in the smartphone has become very popular [18,19]. It can provide users with positioning, navigation, and timing (PNT) services, which brings great convenience to our daily life. Generally, GPS L1 centered around 1.575 GHz is the most commonly used band. Moreover, due to the radiation direction of mobile antenna often changing with the position of the user's hand and the orientation of the mobile phone, the linearly polarized GPS antenna is also widely used in mobile terminal. Furthermore, to cover more desired operating bands like GPS band, numerous techniques such as match network are being studied extensively. In reference [20], a wideband matching network embedded in the feeding strip is used to generate extra resonances, which greatly widens the antenna's lower and higher bands.

In this paper, a dual-antenna system for metal-rimmed smartphone applications is proposed. This dual-antenna system consists of two inverted-F antennas sharing a common part of metal rim. By introducing two gaps in the outer metal rim and a middle shorting branch between two IFAs, the isolation can be improved to some degree. A GPS band can be generated by a matching network embedded in the feeding line of one antenna. Finally, a dual-antenna system in which one antenna covers GPS and WIFI bands while the other one covers WIFI band, can be realized. The detailed radiation mechanism and diversity performance of this dual-antenna system are also presented.

## 2. Antenna Configuration

Figure 1 shows the geometry and detailed dimensions of the proposed dual-antenna system. A 0.8-mm thick FR4 substrate ($\varepsilon r = 4.4$, $\tan\delta = 0.024$), is used as the system circuit board, and its size of $155 \times 75$ mm$^2$ is a typical 5.5-in smart terminal size. The system ground is mounted on the back side of the substrate which is surrounded by a metal rim with two symmetrical gaps of 1mm. The metal rim has a height of 6 mm and a thickness of 0.3 mm. To reduce the influences of the touch screen of the smartphone on antenna radiation performances, the antenna is placed along the short edge of the system ground plane. Moreover, the height of ground clearance is only 4.5mm, which can no doubt satisfy the requirements of a larger display and narrower frame for terminal devices. As can be seen from Figure 1b, this dual-antenna system contains two folded inverted-F antennas (IFAs) with a similar structure, namely, Ant1 and Ant2, which both include a feeding line, a shorting branch and a part of a metal rim. However, in order to achieve good impedance matching over the entire operating bands, Ant1 has a shorter horizontal distance between the feeding line and shorting branch. A matching circuit including two chip inductors (L1 = 6.2 nH and L2 = 2.7 nH) and a chip capacitor (C1 = 1 pF) are located in the feeding line of Ant1, providing an extra resonant frequency at 1.575 GHz. The two IFAs are connected together along their horizontal strips back-to-back. The middle shoring branch HL connects the system ground and the metal rim, which can effectively improve the isolation between two IFAs. Ports 1 and 2 are conducted to excite the inverted-F antennas via 50 Ω mini coaxial feedlines.

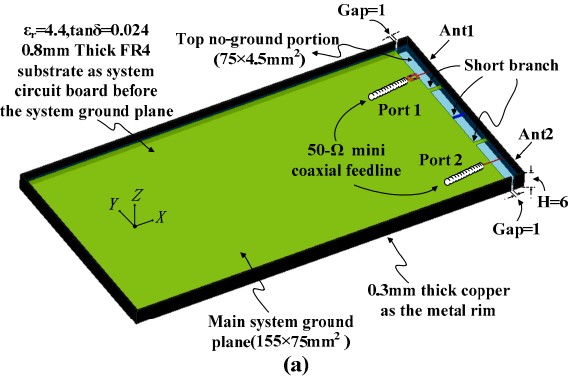

**Figure 1.** *Cont.*

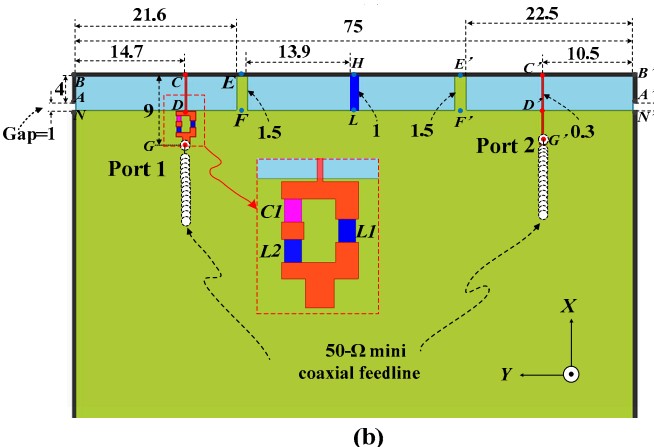

**Figure 1.** The dual-antenna system configuration: (**a**) geometry; (**b**) detailed dimensions (in mm).

## 3. Working Principles

The inverted-F antenna evolves from the basic quarter-wave monopole antenna, thereby their operating principles are similar. Moreover, because of the IFA's low profile characteristics, the IFA is widely used in mobile terminals. This paper proposes that two IFAs share a common metal strip functioned as their radiating arm. The total radiation branch lengths (AB + BE + EF) of Ant1 and (A′B′ + B′E′ + E′F′) of Ant2 are 30.1 mm and 31 mm, respectively, corresponding to a one-quarter wavelength of the center working frequency at 2044 MHz in free space. A high frequency structure simulator (HFSS) is used for simulation. To understand more thoroughly the principle of the proposed dual-antenna system, two reference antennas (Ref-1 and Ref-2) are introduced. Figure 2a shows the simulated S-parameter curve of Ant1 for the proposed antenna and Ref-1 without the matching network. It can obviously be seen that Ant1 of the proposed antenna with matching network generates a new resonant frequency at 1.575 GHz. Figure 2b shows the simulated S parameter of the proposed antenna and Ref-2 without the center shorting branch. As can be seen, the mutual coupling parameter (S21) of the proposed dual-antenna system is less than that of Ref-2 over the GPS and WIFI bands. To better understand the reason for this, the simulated surface current distributions for the proposed antenna and Ref-2 are shown in Figure 3. It can be seen that their surface current distributions are similar in general. However for the proposed antenna, the current is stronger in the shorting branch HL and the center zero current area is obviously larger than that of Ref-2. Thus, this indicates that the isolation can be improved by introducing the shorting branch between the metal rim and the ground plane.

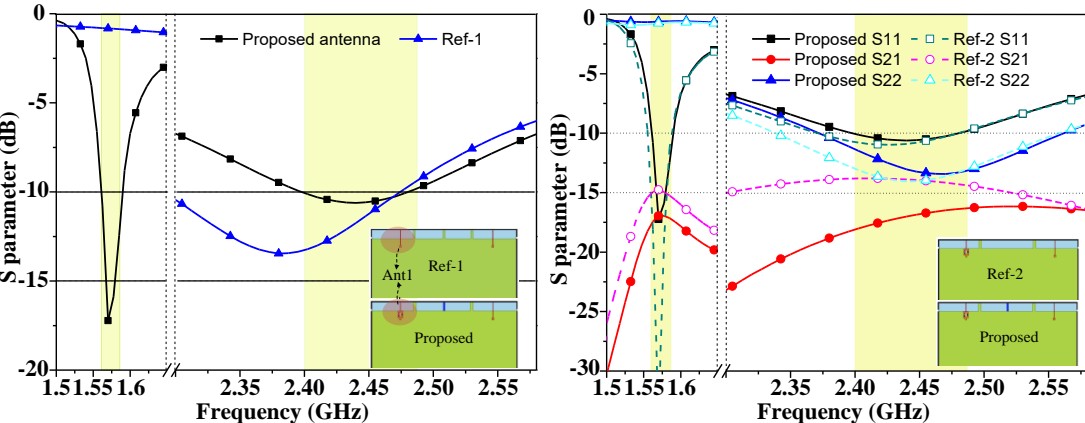

**Figure 2.** Simulated S parameter of Ant1 for the proposed antenna with and without (**a**) the matching network (Ref-1) and (**b**) the middle shorting strip (Ref-2).

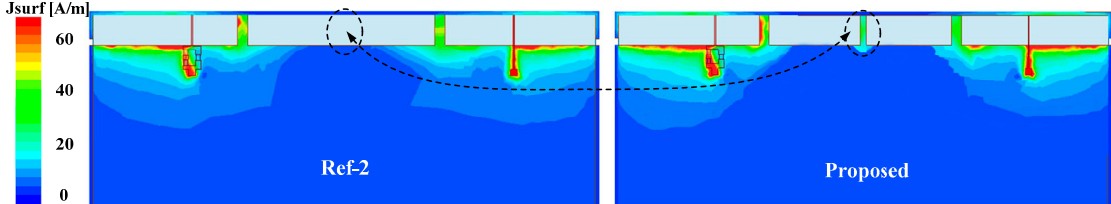

**Figure 3.** Simulated surface current distributions for proposed antenna with the center shorting strip and the reference antenna but without the middle shorting strip.

Figure 4 shows the effect of different values of L1, L2 and C1 on resonant frequencies, respectively. It can be seen that the values of L1, L2 and C1 obviously influence resonant frequencies. When L1 increases from 5.9 nH to 6.5 nH, as shown in Figure 4a, the resonant frequency decreases. Similarly, as the values of L2 and C1 increase, resonant frequencies tend to move toward lower bands. To achieve demanded frequency responses during design process, the values of L1, L2 and C1 in the matching network ought to be controlled and tuned together.

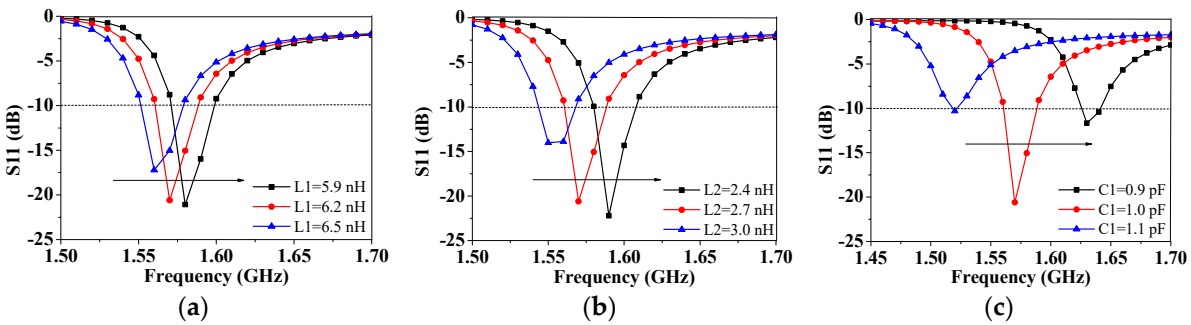

**Figure 4.** Simulated S11 with different values of the matching network: (**a**) L1, (**b**) L2 and (**c**) C1.

To further verify the proposed antenna owning an excellent performance regarding stability, the S parameter with different widths of the shorting branch Ws for the proposed and Ref-2 antennas are shown in Figure 5, while other parameters of the two antennas remain the same. As can be seen from Figure 5a, for Ref-2, when Ws increases from 1 mm to 5 mm, the resonant frequency especially in WIFI band changes dramatically and the isolation in GPS and WIFI bands varies obviously as well. For the proposed antenna, however, the return loss and transmission coefficient curve shown in Figure 5b barely changes. According to the analysis above, it can be seen that the structure of the proposed dual-antenna system is suitable for practical application.

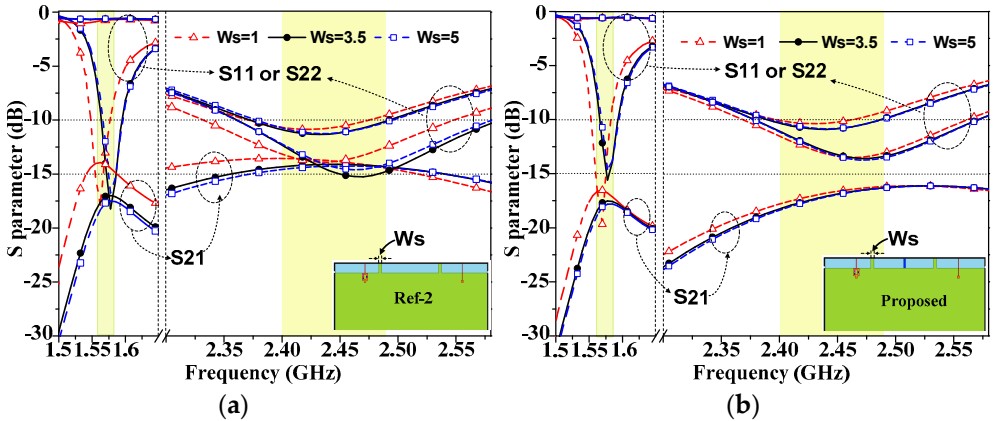

**Figure 5.** Simulated S parameter as a function of Ws, the width of shorting branch for (**a**) the reference antenna (Ref-2) and (**b**) the proposed antenna.

Figure 6a shows the effects of gap width on the antenna performance. When the gap width varies from 0.75 mm to 1.25 mm, the resonant frequency in WIFI band of the proposed antenna is shifted to a higher frequency. In addition, the port matching becomes better and the bandwidth becomes wider, meanwhile the isolation remains lower than −15 dB. As shown in Figure 3, the current is weak in the end part of the radiation branch for IFA, nevertheless the corresponding electric field is strong, which explains why the width of gap has great impact on the antenna performance. Although the wide metal gap is beneficial to the radiation of electric field energy, a too wide gap will damage the esthetic appearance of the smartphone. Thus, the compromising width of 1 mm is chosen, which not only can maintain the aesthetic look of the smartphone, but also meet the desired performance of the proposed dual-antenna system. Furthermore, because some components such as the camera, microphone and USB are normally placed in the antenna ground clearance for smartphone, the stability of the antenna is another important criterion to evaluate its performance. It is demonstrated that the middle shorting branch not only can improve the isolation of the MIMO antenna, but also can enhance its stability. An antenna with USB is simulated and analyzed based on the proposed antenna. As shown in Figure 6b, when a metallic USB connector is placed on the middle shorting branch of the proposed antenna, all the curves of S parameter are almost unchanged. These results show that the metallic USB connector has little effect on the antenna performance.

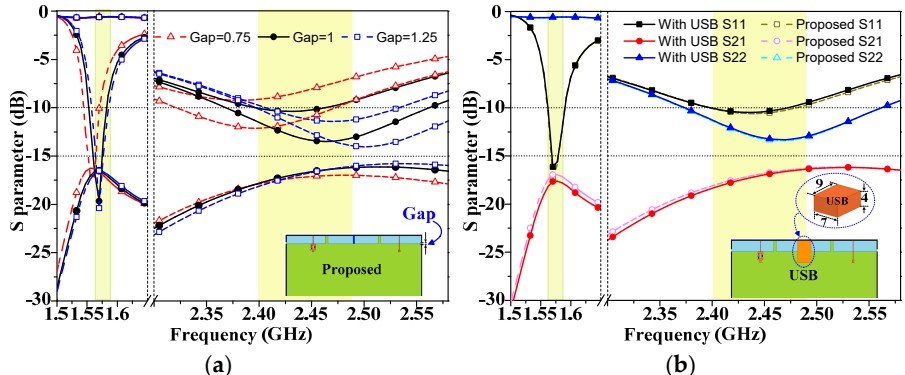

**Figure 6.** Simulated S parameter for the proposed antenna (**a**) as a function of Gap width, and (**b**) with a USB connector.

The effects of the user's hand on the antenna performances are also studied. Figure 7 shows the simulated S-parameters and efficiency at single-hand operation mode, in which the hand is in close proximity to the bottom section of the smartphone device. As can be seen, a slight impedance mismatch appears at the antenna. The antenna can retain good radiating abilities with desirable antenna performances.

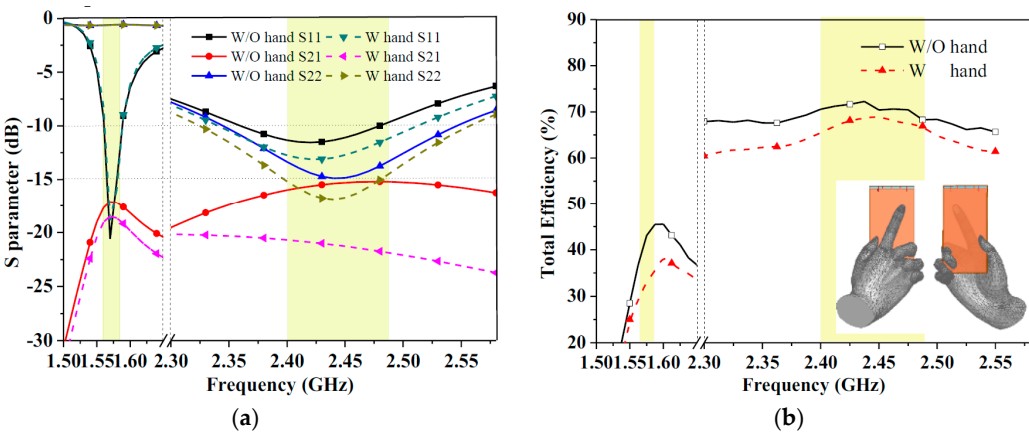

**Figure 7.** Simulated results with hand effect on (**a**) S-parameters and (**b**) Antenna efficiencies.

## 4. Measurement and Antenna Performance

The proposed dual-antenna system is fabricated and its front and back views are shown in Figure 8. It can be seen that the metal rim of this prototype is connected with the ground plane through copper foil. To excite the dual-antenna system, the outer conductors of two 50 Ω coaxial cables are welded to the system ground, and their inner conductors via the dielectric substrate are fixed to the feeding ports. The simulated and measured S parameter of this dual-antenna system are shown in Figure 9 with good agreement, when one port is excited and the other port is terminated to a 50 Ω load. Some discrepancies exist due to manufacturing errors and differences in the impedance between the inter conductor of coaxial wire and the narrow feedline. From this figure, it can be learned that the measured S11 curve of Ant1 is less than −10 dB over the GPS and WIFI bands, while the measured S22 curve of Ant2 is lower than −10 dB over the WIFI band. Meanwhile, the measured isolation between two IFAs of GPS and WIFI bands is larger than 22 dB and 19 dB, respectively, which well meets the design requirement of this dual-antenna system. The SATIMO microwave anechoic chamber is used for radiation measurements of this dual-antenna system. Figure 10 shows the measured efficiency and gain of the proposed antenna. For Ant1, the measured efficiencies are about 37.1–45.5% and 69.3–73.1% in GPS and WIFI bands, respectively. For Ant2, the measured efficiency is about 71.5–77.2% in the WIFI band. The measured gains of Ant1 ranges from 1.0 to 2.1 dBi in the GPS band and from 5.0 to 5.1 dBi in the WIFI band. The measured gain of Ant2 ranges from 5.1 to 5.5 dBi in the WIFI band.

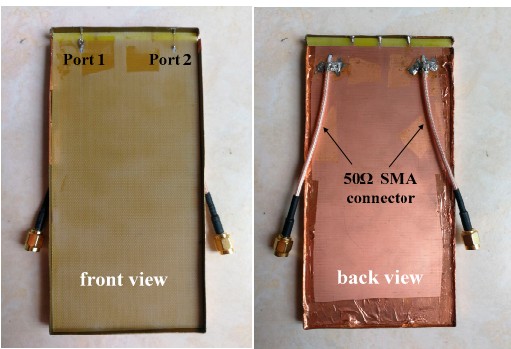

**Figure 8.** The photos of the fabricated dual antenna system.

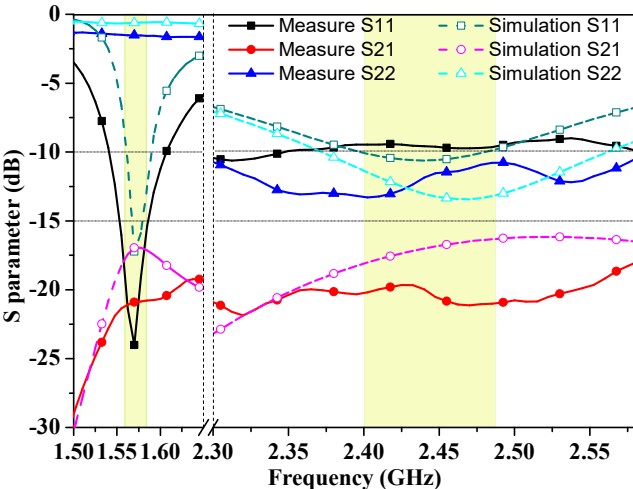

**Figure 9.** Simulated and measured S parameter of the proposed antenna.

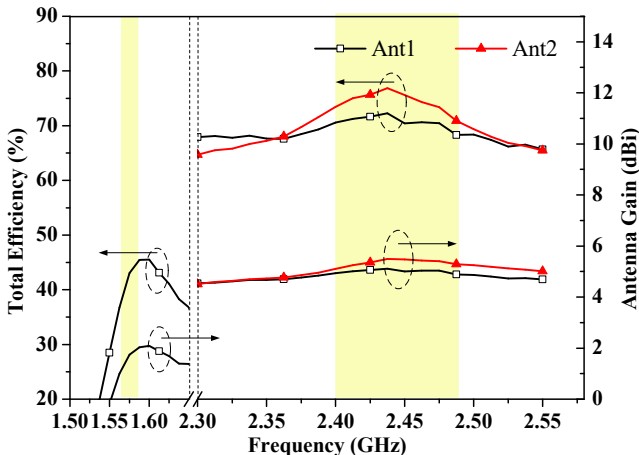

**Figure 10.** Measured antenna efficiency and gain for Ant1 and Ant2.

The measured radiation patterns in the three principle planes for Ant1 and Ant2 at 2450 MHz are plotted in Figure 11. As can be seen, for YOZ and XOZ planes, Etheta and Ephi radiations between two IFAs are complementary. For XOY plane, Ephi radiation between two IFAs is complementary. Therefore, the radiation patterns of the proposed dual-antenna system demonstrate far-field complementary characteristics for the two antennas at 2450 MHz, which helps to reduce the values of ECC over the WIFI bands. Figure 12 shows the measured 2 dimensional radiation patterns for Ant1 at 1570 MHz. The ECC and MEG calculated from the measured complex E-field patterns of the manufactured dual-antenna system are summarized in Table 1. As can be seen, the ECC values are less than 0.02 over the WIFI bands, which means low correlation and good diversity performance are achieved. Moreover, the proposed antenna can achieve |MEG1 − MEG2| < 1 dB within the entire WIFI bands, which indicates that it is suitable for practical smartphone MIMO antenna applications.

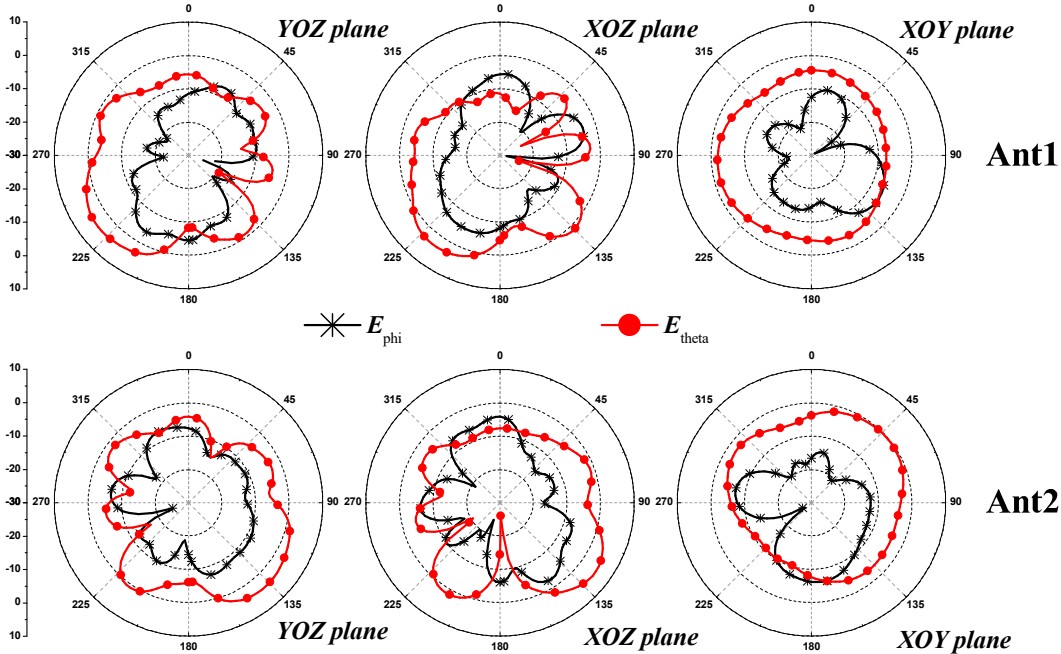

**Figure 11.** Measured 2 dimensional radiation patterns for Ant1 and Ant2 at 2450MHz.

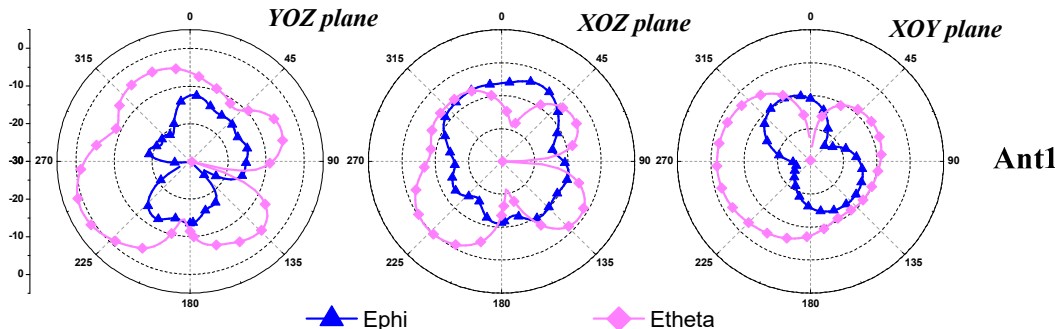

**Figure 12.** Measured 2 dimensional radiation patterns for Ant1 at 1570MHz.

**Table 1.** Diversity performance of the proposed antenna.

| Frequency (MHz) | 2375 | 2400 | 2425 | 2450 | 2475 | 2500 |
|---|---|---|---|---|---|---|
| $\eta_{tot}1$ | 69.8% | 73.5% | 75.6% | 75.6% | 73.4% | 69.3% |
| $\eta_{tot}2$ | 68.5% | 70.5% | 71.7% | 70.4% | 70.5% | 68.4% |
| Peak-gain1 (dB) | 4.86 | 5.10 | 5.35 | 5.47 | 5.40 | 5.24 |
| Peak-gain2 (dB) | 4.77 | 4.94 | 5.06 | 5.00 | 5.04 | 4.87 |
| ECC | 0.0094 | 0.0105 | 0.0113 | 0.0114 | 0.0113 | 0.0106 |
| MEG1 (dB) | −3.73 | −3.51 | −3.39 | −3.41 | −3.56 | −3.83 |
| MEG2 (dB) | −3.88 | −3.75 | −3.68 | −3.78 | −3.81 | −3.96 |

## 5. Conclusions

A dual-antenna system with common metal rim for smartphone applications in GPS and WIFI bands has been designed and analyzed. By introducing two gaps in the metal rim and a middle shorting branch between the metal rim and ground plane, good isolations of better than 22 dB and 19 dB can be achieved in GPS and WIFI bands, respectively. Meanwhile, the good stability of the proposed antenna has been verified. With the aid of a matching network, the extra GPS band for Ant1 is realized. The proposed dual-antenna system has been designed, manufactured and measured. Encouraging results regarding S-parameters, antenna efficiency and gain, ECC, along with MEG are obtained. Thus, the proposed design of a GPS-WIFI-band dual antenna system is very promising for metal-rimmed smartphone applications.

**Author Contributions:** Design and analysis, Z.X. and Q.Z.; simulation and modification, Y.S. and S.H.; revision, measurement and validation, C.D.

**Funding:** This work was supported by National Natural Science Foundation of China (Grant No.: 61301052), State Key Laboratory of Advanced Materials and Electronic Components (Grant No.: 180565) and Sichuan Science and Technology Program (Grant No.: 2017GZ0102, 2018GZ0010, and 2017GZ0020).

**Conflicts of Interest:** The authors declare no conflict of interest.

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
