# Peer review of "A Dual-Band Dual-Antenna System with Common-Metal Rim for Smartphone Applications"

_electronics, doi:10.3390/electronics8030348_

Round 1

Reviewer 1 Report

The paper analyse the use of IFA antennas in cellular phones. Even if no theoretical advances are present, the paper can be useful for practitioners. 

Analysis and measurement is well done. However, the performance of cellular phones can significant change in real applications due to the presence of the hand. The authors should consider this problem. Of course the best choice would be to include some numerical or experimental results. However this reviewer fully understand that this would require a large work. Alternatively, the reviewer suggest to add a discussion on this problem, giving some general indications on the effects of the hand of the user on the radiated field and the efficiency.

Author Response

Thanks for your kind comments. 

Reviewer 2 Report

The paper "A Dual-Band Dual-Antenna System with Common-metal Rim for Smartphone Applications" presents an antenna system that provides good performance in the GPS and WiFi bands even with a metallic rim embedded around the cell phone. The proposed design achieves good performance in both bands and good decoupling.

The paper is well written and the motivation is clear (apparent from recent news descibing iPhones' loss of functionality due to the metallic rim). The problem that I have with this paper is that it describes a work of engineering, not research. The work does not involve any novel technique or design principle; and the problem seems to have been already addressed in commercial products. How does this antenna compare to that of similar smartphones? How is the antenna showing MIMO capability? 

Author Response

Thanks for your kind comments. 

Reviewer 3 Report

The article is a well documented paper with explanation of the design, simulations and measurements.

My only question for improvement is: what is the effect of tolerance and loss of L1, L2 and C1 elements on the input reflection and antenna efficiency.

I would suggest to make simulation using L1, L2 and C1 element values in the tolerance interval and show the effect on resonance. Similarly it can be interesting how the antenna efficiency changing by taking into account the matching element losses.

Author Response

Thanks for your kind comments.

Round 2

Reviewer 2 Report

The authors have partially addressed my comments. I will not oppose to publication.

Author Response

We sincerely appreciate the referee’s insightful comments and constructive input to our work.